# Protection Schemes in HPON Networks Based on the PWFBA Algorithm

**DOI:** 10.3390/s22249885

**Published:** 2022-12-15

**Authors:** Rastislav Róka, Radek Fujdiak, Eva Holasova, Karel Kuchar, Milos Orgon, Jiri Misurec

**Affiliations:** 1Institute of Multimedia Information and Communication Technologies, Slovak University of Technology, Ilkovicova 3, 812 19 Bratislava, Slovakia; 2Department of Telecommunications, Faculty of Electrical Engineering and Communications, Brno University of Technology, Technicka 12, 616 00 Brno, Czech Republic

**Keywords:** optic fiber, telecommunication network, infrastructure, network traffic control, HPON, PWFBA, channel allocation, optical fiber networks, scheduling algorithms, systems simulation, telecommunication services

## Abstract

In this paper, possibilities for network traffic protection in future hybrid passive optical networks are presented, and reasons for realizing and utilizing advanced network traffic protection schemes for various network traffic classes in these networks are analyzed. Next, principles of the Prediction-based Fair Wavelength and Bandwidth Allocation (PFWBA) algorithm are introduced in detail, focusing on the Prediction-based Fair Excessive Bandwidth Reallocation (PFEBR) algorithm with the Early Dynamic Bandwidth Allocation (E-DBA) mechanism and subsequent Dynamic Wavelength Allocation (DWA) scheme. For analyzing various wavelength allocation possibilities in Hybrid Passive Optical Networks (HPON) networks, a simulation program with the enhancement of the PFWBA algorithm is realized. Finally, a comparison of different methods of the wavelength allocation in conjunction with specific network traffic classes is executed for future HPON networks with considered protection schemes. Subsequently, three methods are presented from the viewpoint of HPON network traffic protection possibilities, including a new approach for the wavelength allocation based on network traffic protection assumptions.

## 1. Introduction

Passive Optical Network (PON) is a fiber-based point-to-multipoint optical network communication technology with no active elements in the signals path from source to destination (single optical fiber serves multiple endpoints by using unpowered/passive fiber optic splitters, which divides the fiber bandwidth to them), with main advantages such as [1] longer distances (i.e., as the last mile technology between internet provider and customer—approx. 20 km) [2], higher bandwidth (futuristic view up to 25G-PON to 50G-PON [3]), downstream video broadcasting, eliminated need for additional electronic devices in the network (thanks to its passive characteristic), and easy upgrades to higher bit rates (or additional wavelengths).

Future access technologies utilized in passive optical networks must be able to provide high sustainable bandwidths on a per-user basis while keeping capital and operational expenditures as low as possible. Therefore, Next-Generation Passive Optical Networks (NG-PON) need to provide survivability schemes in a cost-efficient way, whereas the growing importance of uninterrupted internet access makes fault management an important challenge [4]. The reliability requirements may depend on user profiles. Thus, NG-PON networks should also support the end-to-end protection for selected users when requested [5]. Within this context, it is significant to discover an effective way to analyze network traffic protection schemes for utilization in future passive optical networks [6].

We introduce new possibilities for network traffic protection in future Hybrid Passive Optical Networks (HPON) [7], including reasons for using the advanced traffic protection schemes for various network traffic classes in those passive networks. Furthermore, principles of the Prediction-based Fair Wavelength and Bandwidth Allocation (PFWBA) algorithm are explained in detail, with a close focus on a Prediction-based Fair Excessive Bandwidth Reallocation (PFEBR) algorithm, with an Early Dynamic Bandwidth Allocation (E-DBA) mechanism and Dynamic Wavelength Allocation (DWA) scheme. Subsequently, three methods are analyzed and evaluated from the viewpoint of HPON network traffic protection possibilities. The first method presents an original approach to the PFEBR algorithm adapted to network traffic protection schemes. The second method introduces our newly developed approach to wavelength allocation based on network traffic protection assumptions. The third method, based on fixed wavelength priority bandwidth allocation, we accommodated directly to specific network traffic classes.

The paper is structured as follows. Section 2 introduces the finer details of HPON networks and the importance of network traffic protection in such networks. Section 3 provides a detailed description of PFWBA, PFEBR and E-DBA algorithm principles. Section 4 provides details about our simulation for HPON, including the results. Last but not least, Section 5 provides conclusions of our findings.

## 2. Network Traffic Protection in HPON Networks

The HPON presents an intermediate stage in the migration scenario to the fully operational Wavelength Division Multiplexing-Passive Optical Network (WDM-PON) networks, which have a positive future thanks to their ability to satisfy the growing bandwidth demands [8]. Hybrid passive optical networks, except for other characteristics, can utilize both Time Division Multiplex (TDM) and Wavelength-Division Multiplexing (WDM) techniques. It means that various wavelengths can be considered for the network traffic. Thanks to WDM technology, we can efficiently utilize the optical transmission medium, which is very important from the viewpoint of growing bandwidth demands and cheap services. In the present day, there are different standardized passive optical technologies with various dedicated wavelengths able to cooperate. Moreover, the availability of various wavelengths transmission channels leads to the extension of advanced network traffic protection schemes. Therefore, new wavelength allocation methods must be considered in addition to effective bandwidth capacity utilization in HPON networks [9,10].

Future HPON architectures can be proposed without or with network traffic protection schemes. Therefore, they can be designed keeping in mind different possible paths for network deployment and protection upgrades and can be proposed with different levels of network traffic protection. The proposed survivable architectures can also be applied to Time-Division Multiplexing-Passive Optical Networks (TDM-PON) with more than one stage of remote nodes based on power splitters [11,12]. A benefit of the network traffic protection deployed in HPON networks can be obtained as a consequence of the reliability performance improvement and the service interruption decrements experienced by users [13,14,15,16]. It can be beneficial to either provide protection functionalities at the time of the HPON network deployment or at least install a sufficient amount of optical fibers in advance. Furthermore, reasons for network traffic protection are very substantial from the viewpoint of signal transmission. A clear benefit can be shown when network planning is completed with a possible protection upgrade, which leads to a decrease in investment costs. This confirms the importance of the right deployment plan for future hybrid passive optical networks.

HPON networks present a base for converged Fibre-Wireless Passive Optical Networks (Fi-Wi PON) [17]. For that reason, it is necessary to provide a certain level of network traffic protection and restoration [8,11,12]. Different protection schemes can be proposed, starting from a no-protection scenario towards proposed architectures with the protection. However, in a case that no hardware protection is utilized at the network creation, appropriate network traffic protection schemes can be realized. Thus, transmission channels provided with various wavelengths can be involved in providing protection paths for high-priority network traffic classes.

Except for optimization of the optical power budget for passive optical network designs [18], the network traffic protection securing and the system recovery in the case of failures is one of the most important issues in HPON networks. As future passive optical networks transmit aggregated high-speed data from up to hundreds of customers, network units and/or distribution failures represent a serious problem. Research on fault-tolerant HPON topologies recommends the utilization of duplicated optical fibers between the Optical Line Terminal (OLT) and Optical Network Units (ONU), duplicated optical components and the supplementary circuit included in the Remote Node (RN) in one common entity [16] even though these schemes are non-adequate due to their high redundancy with significant expenses [13,19,20]. Except for hardware realization, costlier effective solutions that focus on dynamic wavelength and bandwidth allocation algorithms can be utilized for enhancing differentiated network traffic protection schemes for different traffic classes [21,22].

In this paper, the attention is focused on possible advanced network traffic protection in HPON networks utilizing an appropriate enhancement of the PFWBA activity. In Section 3, principles of the PFWBA algorithm are introduced in detail, focusing on the Prediction-based Fair Excessive Bandwidth Reallocation (PFEBR) algorithm with the Early Dynamic Bandwidth Allocation (E-DBA) mechanism and the related Dynamic Wavelength Allocation (DWA) scheme. This PFWBA algorithm includes the PFEBR and the DWA scheme. These two algorithms are not executed simultaneously, but first, the OLT terminal calculates the time intervals for individual ONU units, and subsequently, working and/or protection wavelengths can be assigned to these calculated time intervals. To analyze various network traffic protection schemes utilizing the wavelength allocation cases possible in HPON networks, a simulation program with the enhancement of the PWFBA algorithm is realized, and an evaluation of different network traffic protection types available for future HPON networks is executed in Section 4. Particular requirements are more or less important for the selected HPON network topology and are satisfied by the presented enhancement of the PFWBA algorithm.

## 3. PFWBA Algorithm Principles

### 3.1. E-DBA and PFEBR Principles

The robust PFWBA contains DWA schemes and the E-DBA mechanism for the PFEBR. In the standard Dynamic Bandwidth Allocation (DBA) scheme, the OLT unit starts a process of bandwidth allocation based on time intervals after receiving *REPORT* messages from all ONU units. The early DBA mechanism (E-DBA) ensures the sequence of transmitting *REPORT* messages into the OLT by delaying ONU units with unstable network traffic based on the *B_V_*, which is represented as the ONU group with a higher variance than the mean variance.

The E-DBA mechanism consists of two operations. In the first step, the OLT executes the DBA scheme after receiving *REPORT* messages received at the end of ONU_i−1_. This operation reduces the idle period in the standard DBA algorithm and obtains actual information for ONU units with the unstable network traffic, leading to improving the prediction accuracy in the next service cycle. In the second step, a time interval is assigned to each ONU unit based on the network traffic variance of all ONU units in decreasing order, and at the same time, the *B_V_* group is updated by adding some unstable network traffic ONU units with higher variances. This operation mitigates a variance by shortening the waiting time before transmitting data from unstable network traffic ONU units [19].

The PFEBR algorithm calculates a variance of each ONU unit based on previous network traffic information and sorts variances in decreasing order. In this way, an unstable degree list is acquired. A calculation of the variance *V_i_* for the ONU_i_ unit can be expressed as: (1)$$V_{i}=\frac{1}{N_{H}}\sum_{n\in PC}{\left(B_{i,n}^{Total}-B_{i}^{Mean}\right)}^{2}$$
where *B^Total^* represents the sum of differentiated network traffic classes (Assured Forwarding (*AF*), Best Effort (*BE*) and Expedited Forwarding (*EF*)) of the ONU_i_ unit in the given (*n*) cycle ∈ PC (Previous Cycle), *B^Mean^* is the mean of *B^Total^* values and *N_H_* presents the number of historical *REPORT* messages. The *B_V_* group contains ONU units with a higher variance than the mean variance *V^Mean^* calculated as:(2)$$V^{Mean}=\frac{1}{N}\sum_{i=1}^{N}V_{i}$$
where *N* represents a number of ONU units in the system. The E-DBA mechanism moves *REPORT* messages from the *B_V_* group between previous and current ONU units. The PFEBR algorithm requests actual information from the unstable network traffic ONU list to avoid prediction inexactness.

After sending data from all ONU units based on the unstable degree list (UDL), the PFEBR predicts the future network traffic requirement based on the bandwidth request according to this UDL. Predicted requests *R^C^* based on differentiated network traffic classes for all ONU units are defined as follows:(3)$$R_{i,n}^{EF}=B_{i,n}^{EF}$$
(4)$$R_{i,n+1}^{C}=\left(1+\alpha \right)\cdot B_{i,n}^{C}$$
where *B^C^* presents the requested bandwidth of the ONU_i_ in the given (*n*) cycle for differentiated network traffic classes *C* ∈{AF,BE} and α means the linear estimated credit [19].

The PFEBR algorithm executes the Excessive Bandwidth Reallocation (EBR) process after finishing the bandwidth prediction needed for each ONU unit. The PFEBR scheme can provide a fairness approach to the EBR in accordance with the guaranteed bandwidth. First, the fair EBR operation in the PFEBR must calculate the *R^Total^* for each ONU unit. The size of the available bandwidth can be expressed as:(5)$$B^{Available}=C_{cap}\cdot \left(T_{cycle}-N\cdot g-N_{V}\cdot g\right)-N\cdot L_{CM}$$
where *C_cap_* is the OLT line capacity in bit/s, *T_cycle_* presents the maximum cycle interval, *g* is the guard time, *N* is the number of ONU units, *N_V_* is the number of ONU units in the *B_V_* group, and the control message length *L_CM_* is 512 bits (64 octets). The ONU_i_ unit with the maximum residual bandwidth is then selected from unallocated ONU units. The granted bandwidth allocated for the ONU_i_ unit $G_{i,n}^{Total}$ in the next cycle is defined as:(6)$$G_{i,n}^{Total}=min\left(B^{Available}\cdot \frac{S_{i}}{\sum_{k\in UN}S_{k}},R_{i,n}^{Total}\right)$$
where *R^Total^* is the sum of differentiated network traffic loading after prediction from the ONU_i_ unit in the given (*n*) cycle, *S_i_/∑ S_k_*, where *k*∈ *UN* (unallocated) is the ratio of the available bandwidth *B^Available^* assigned to the ONU_i_ unit. The granted bandwidth for particular differentiated (*EF*, *AF* and *BE*) network traffic classes is as follows:(7)$$G_{i,n+1}^{EF}=R_{i,n}^{EF}$$
(8)$$G_{i,n+1}^{AF}=min\left(G_{i,n+1}^{Total}-G_{i,n+1}^{EF},R_{i,n}^{AF}\right)$$
(9)$$G_{i,n+1}^{BE}=G_{i,n+1}^{Total}-G_{i,n+1}^{EF}-G_{i,n+1}^{AF}$$

This process continues until the bandwidth for each ONU unit is allocated. Finally, the PFEBR organizes a broadcasting sequence and a report time for each ONU unit based on the unstable degree list [19].

### 3.2. PFWBA Principles

Cooperation between the DWA algorithm and the PFEBR scheme for enhancement of the system performance can be considered in this way. Before wavelengths are allocated, the PFEBR based on requests from all ONU units determines the transmitting time for the given ONU unit in the current cycle. The PFWBA considers the unstable degree list and enhances the prediction accuracy when scheduling the transmitting sequence after collecting *REPORT* messages from all ONU units. First, the PFWBA scheme divides all ONU units into three groups based on the variance of all ONU units:group 1: if ONU_i_ $\in B_{V}$group 2: *V_i_* > *V^Mean^* and ONU_i_ $\notin B_{V}$group 3: otherwise

Then, the PFWBA scheme assigns the wavelength for each ONU unit gradually group by group. The PFWBA defines two basic variables—the Channel Available Time (CAT) presents the wavelength availability for broadcasting after the time expiration *t*, and the Round Trip Time (RTT) presents the time needed for signal transmission from the OLT terminal to ONU units and back. The PFWBA selects requested time intervals in the same group with a minimum transmission time. This scheduling process representing the DWA is concretely described in [19].

## 4. Simulation Program of the Advanced HPON Network Traffic Protection

We prepared and realized a simulation program in the Java Runtime Environment (JRE) programming software using the Eclipse Integrated Development Environment (IDE) framework that can simulate stochastic network traffic in the HPON network and can be utilized for providing traffic protection paths for different network traffic classes [23]. This program presents activities of the basic PFWBA algorithm and its modifications from the viewpoint of network traffic protection schemes. Specifically, the program presents the dynamic bandwidth and wavelength allocation process in each service cycle managed by the OLT unit. The program also incorporates a generator of the stochastic network traffic with a generation of particular requests for differentiated network traffic classes that are present in the *REPORT* message for individual ONU units. The simulation program works in cooperation with the Adobe Integrated Runtime (Adobe AIR) multiplatform runtime system due to simpler programming of the graphical interface in the program, where many input parameters can be predetermined, for example, the cycle duration, the guard time, the OLT capacity, the number of ONU units, the number of ONU units entering the unstable degree list, the number of wavelengths and the selected method for the wavelength allocation. There is a possibility for step-by-step operation during the bandwidth and wavelength allocation process. Default input parameters at the program initialization are presented in Table 1.

First three values of input parameters are determined by the OLT terminal at the beginning of the network traffic transmission. They allow for the distribution of time slots without network collisions. They are also unchanged until the HPON network architecture is changed. In this case, these values must be recalculated. As an example, if the specific 1 Gbps line capacity per 1 wavelength is supposed, a value of the available bandwidth after subtraction of guard times and *REPORT* messages can be calculated by using Equation (Equation 5). *B^Available^* is the maximum number of bytes that can be assigned in one cycle. If this number is multiplied by the number of cycles per second, then the maximum available capacity can be determined. In the next step, the variance (the difference between the current and previous requests) according to Equation (Equation 1) must be considered.

Therefore, the median for all ONU units is compared with the mean variance *V^Mean^*. The first group of ONU units with the largest variance are entered into the unstable degree list, which means that the PFEBR algorithm predicts their large request changes and accommodates them first. The second group of elements is created by ONU units with variance larger than the median, but they are not included in the unstable degree list. The third group is created with ONU units with variance smaller than the median, and thus, the PFEBR algorithm supposes that these units will have no large change in requests. Therefore, data are assigned last to this group in decreasing order. In addition, the PFEBR algorithm ensures fair bandwidth allocations. If more ONU units are included in the UDL, the PFEBR will be less fair.

Because the PFEBR algorithm belongs to the one-level prediction techniques, it predicts the bandwidth allocated for particular ONU units. The prediction is based on the linear credit as a ratio of the request in the given (*n*) cycle to the total transmitting time of all ONU units in the previous (*n* − 1) cycle. Therefore, the prediction for a given network traffic class is larger than the requested transmission for each ONU unit. In the final step, the bandwidth for ONU units is allocated based on Equations (Equation 7)–(Equation 9).

The bandwidth allocation is realized in groups, i.e., first time intervals are allocated to ONU units from the *B_V_* group and, finally, up to ONU units with a smaller variance. When the OLT unit assigns time intervals to ONU units, it must also assign wavelengths. Without knowing assigned time intervals, an allocation of wavelengths could be very ineffective. Therefore, the PFEBR algorithm is performed before the DWA algorithm. Three methods for the wavelength allocation are analyzed.

### 4.1. Method 1—Uniform Utilization of Wavelengths

The first method for advanced network traffic protection is characterized by a uniform utilization of all possible wavelengths. Its disadvantage is the fact that transmitters and receivers paired with considered wavelengths must be turned on in all ONU units utilized in the HPON network. On the contrary, its advantage is the computing simplicity. For specific network traffic classes, two wavelengths are simultaneously utilized for network traffic protection. In this case, both wavelengths are utilized as working and protection paths. In addition, only half of the transmission capacity can be practically utilized. The measurement data based on method 1 are shown in Table 2. The wavelength allocation using method 1 is shown in Figure 1 (Number of Wavelengths: 2; Available Bandwidth: 972,952 Mbps; 243,238 Bytes/cycle; Total Grant: 192,361 Bytes; Throughput: 39%).

### 4.2. Method 2—Non-Uniform Utilization of Wavelengths

The second method for advanced network traffic protection is trying to utilize only one wavelength, and the second one is not activated until network traffic exceeds the line capacity for the wavelength. This is carried out by a subtraction of transmitting times of particular ONU units from the CAT parameter. A disadvantage of our proposed method is higher computing intensity than in method 1. The measurement data based on method 2 are shown in Table 3. The wavelength allocation using method 2 is shown in Figure 2 (Number of Wavelengths: 2; Available Bandwidth: 972,952 Mbps; 243,238 Bytes/cycle; Total Grant: 138,226 Bytes; Throughput: 28%). For specific network traffic classes, the first wavelength is realized as the working path; the second one is considered as the protection path. Its great advantage is the power saving if the second wavelength is not utilized at that moment. Therefore, except from realizing advanced network traffic protection, possible power savings in ONU units can be seriously considered. In praxis, this power saving can be increased with a higher number of ONU units in HPON networks.

### 4.3. Method 3—Fixed Wavelength Priority Bandwidth Allocation FWBPA

This method can be applied in applications very sensitive to delay. Because the third method for advanced network traffic protection is working with three different wavelengths, each network traffic class has its own dedicated wavelength. In this way, a small total delay is ensured. A disadvantage is the data stream separation from the ONU unit, whereby this data stream is commonly transmitted in methods 1 and 2. Moreover, method 3 uses the RTT parameter for each network traffic class separately, which means three more times. Then, the method FWBPA is not so effective as method 2, but it allows for the optimization of packet losses and the delay in the AF network traffic dominating in access networks in the present day [20]. The wavelength allocation using method 3 is shown in Figure 3 (Number of Wavelengths: 3; Available Bandwidth: 972,952 Mbps; 243,238 Bytes/cycle; Total Grant: 459,012 Bytes; Throughput: 62%).

For specific network traffic classes, three wavelengths are simultaneously utilized. Each wavelength is realized as the working path for a specific network traffic class and simultaneously as the protection path for the other two network traffic classes in the case of wavelength failure. If the channel capacity is sufficient, the allocated bandwidth is always higher than the requested bandwidth. This is caused by the linear credit that is always positive. As the PFWBA algorithm supports Quality of Service (QoS) requirements, the EF and AF network traffic classes sensitive to delay are preferred to the non-sensitive BE network traffic class. Data not transmitted in the current cycle will be included in the next cycle and will be transmitted if the channel capacity is sufficient. Generally, this BE network traffic does not require any specifications for the bandwidth guarantee and transmission delay.

## 5. Conclusions

In this paper, the enhancement of HPON network traffic protection schemes based on basic features of the PFWBA algorithm is analyzed. The PFWBA algorithm based on prediction allows for a dynamic and effective utilization of network transmission capacities. Therefore, our selected PFWBA algorithm belongs to algorithms that could possibly be implemented for network traffic protection in hybrid passive optical networks. Using a realized simulation program, three methods were analyzed and evaluated from the viewpoint of traffic protection possibilities. The first method presents an original approach of the PFEBR algorithm adapted to network traffic protection schemes. The second method introduces a new approach for the wavelength allocation based on network traffic protection assumptions. The third method based on fixed wavelength priority bandwidth allocation is accommodated directly to specific network traffic classes.

For advanced network traffic protection schemes in HPON networks, two or three wavelengths can be simultaneously utilized. There exist two basic approaches to utilizing wavelength allocation. In the first approach, the same wavelength can be realized as the working and protection path for specific network traffic classes (a method with uniform utilization of wavelengths and the FWBPA method). In the second approach, different wavelengths are utilized as the working and protection paths for all network traffic classes (a method with the non-uniform utilization of wavelengths). Results from the simulation program show that our proposed method with the non-uniform utilization of wavelengths seems to be more effective from the viewpoint of network traffic protection compared with other considered methods.

Based on the simulation results obtained, we intend to implement the presented methods of wavelength allocation in real HPON systems with considered traffic protection schemes. In some scenarios, each ONU can also process differentiated network traffic classes by varying the number of resources for a defined number of wavelengths. Therefore, the effect of using a different number of resources requested by different traffic classes will be considered in future works.

## Figures and Tables

**Figure 1 sensors-22-09885-f001:**
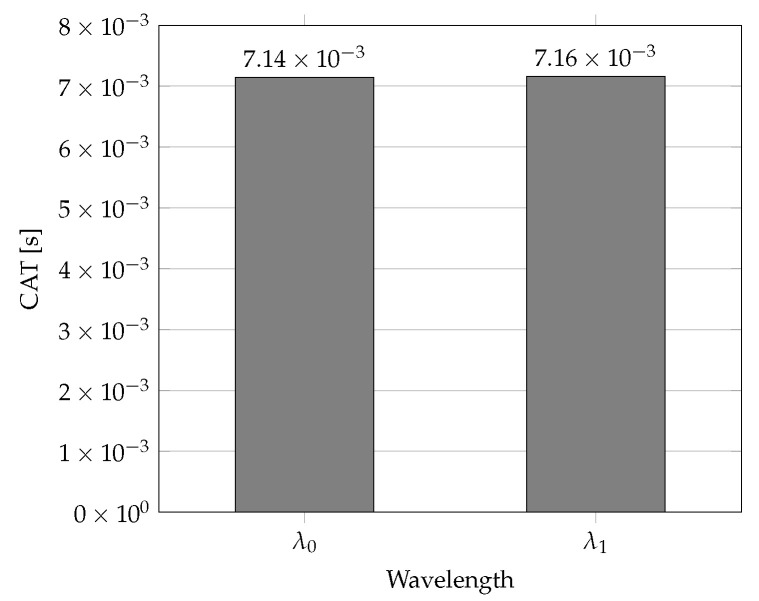
CAT parameter values in s for wavelengths in cycle No. 10 using Method 1.

**Figure 2 sensors-22-09885-f002:**
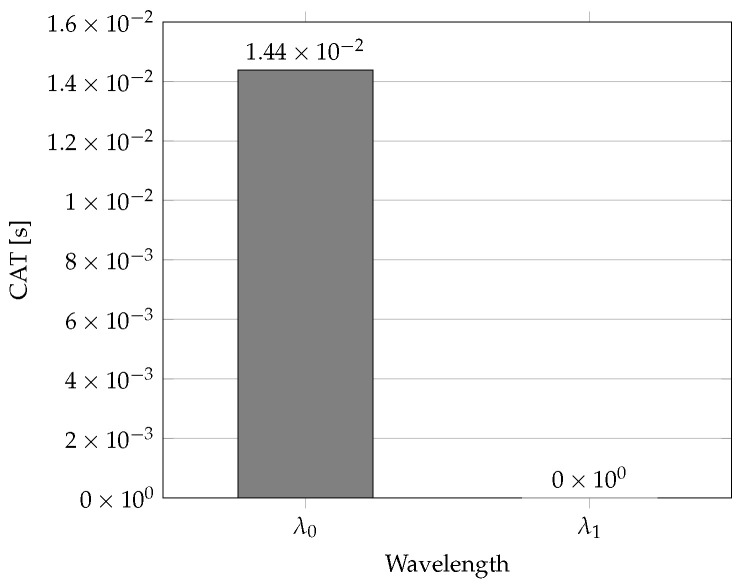
CAT parameter values in s for wavelengths in cycle No. 27 using Method 2.

**Figure 3 sensors-22-09885-f003:**
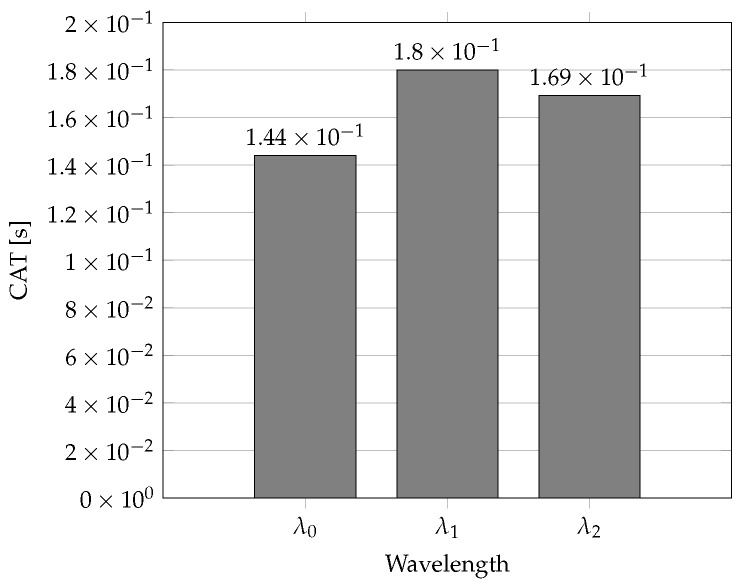
CAT parameter values in s for wavelengths in cycle No. 100 using Method 3.

**Table 1 sensors-22-09885-t001:** Default input parameters in the advanced HPON simulation program.

Parameter	Parameter Value
Service cycle duration	2 ms
Guard time	5 μs
RTT range	50–200 μs
OLT capacity per 1 wavelength	1 Gbps
Number of ONU units	8
Number of units in the UDL	2
Number of ONU wavelengths	2 (method 1 and 2) 3 (method 3)

**Table 2 sensors-22-09885-t002:** Example of the wavelength allocation for actual data transmission demands in bytes from 8 ONUs in cycle No. 10 using Method 1.

ONU No.	Median	EF_pred.	AF_pred.	BE_pred.	EF_grant	AF_grant	BE_grant	Total Granted	Wavelength No.
0	2,347,024	0	4102	9995	0	4102	9995	14,097	0
1	8,952,064	1451	3530	8172	1451	3530	8172	13,153	1
2	19,351,201	5815	17,645	10,351	5815	17,645	10,351	33,811	0
3	55,532,304	4265	16,029	7162	4265	16,029	7162	27,456	0
4	33,031,001	6368	10,869	26,075	6368	18,069	26,075	50,512	1
5	196,077,476	1595	34,140	4363	1595	34,140	4363	40,098	0
6	864,900	505	2041	8482	505	2041	8482	11,028	1
7	92,679,129	745	0	1461	745	0	1461	2206	1

**Table 3 sensors-22-09885-t003:** Example of the wavelength allocation for actual data transmission demands in bytes from 8 ONUs in cycle No. 27 using Method 2.

ONU No.	Median	EF_pred.	AF_pred.	BE_pred.	EF_grant	AF_grant	BE_grant	Total Granted	Wavelength No.
0	58,064,400	2119	0	5501	2119	0	5501	7620	0
1	6,948,496	5148	0	5141	5148	0	5141	10,289	0
2	274,576	8804	19,971	0	8804	19971	0	28,775	0
3	30,140,100	5037	23,569	4393	5037	23,569	4393	32,999	0
4	75,012,921	0	22,934	10,713	0	22,934	10,713	33,647	0
5	16,297,369	2088	6729	5312	2088	6729	5312	14,129	0
6	1,092,025	4560	0	2045	4560	0	2045	6605	0
7	31,329	4162	0	0	4162	0	0	4162	0

## Data Availability

Not applicable.

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
