# Peer review of "Protection Schemes in HPON Networks Based on the PWFBA Algorithm"

_sensors, 2022, doi:10.3390/s22249885_

Round 1

Reviewer 1 Report

This paper presents the analysis of the network traffic protection in future hybrid passive optical networks and the reasons for realizing and utilizing of advanced network traffic protection schemes for various network traffic classes. The Prediction-based Fair Wavelength and Bandwidth Allocation (PFWBA) algorithm is investigated and three methods of the wavelength allocation in conjunction with specific network traffic classes are studied.

The following should be addressed properly before it can be recommended to be published on Sensors.

(1) The novelty and efficiency of the PFWBA algorithm should be pointed out clearly.

(2) In Section 4, the comparison of different methods of the wavelength allocation in conjunction with specific network traffic classes is executed. However, the advantages and disadvantages of the three method are the common sense for the researches.

Author Response

Thank you for your review and please see our response in attached document.

Sincerely yours,
Radek Fujdiak.

Reviewer 2 Report

Authors present possibilities for the network traffic protection in future hybrid passive optical networks. They analyzed reasons for realizing and utilizing of advanced network traffic protection schemes for various network traffic classes in these networks. Authors also realized a simulation program with the enhancement of the PFWBA algorithm which analyzes various wavelength allocation possibilities in (HPON) networks. They also compared different methods of the wavelength allocation in conjunction with specific network traffic classes in the future HPON networks with considered protection schemes.

The article was written correctly and it raises a very interesting problem. 

I have only a few small remarks that do not detract from the significance of the presented results:

- The first sentence in the Introduction. Should be Passive Optical Network (singular).

- More details about the simulator would be useful, such as the programming language in which it was written, computational complexity or program execution time. The authors mention only Adobe Air.

- The data in tables 2-4 should be better explained. Units are missing.

- In traffic theory, classes are also defined by the number of requested resources (e.g. throughput). Would a different number of resources requested by different traffic classes affect the results obtained?

- Will the authors intend to compare the obtained simulation results with real systems in further works?

- The article does not clearly state whether the authors use algorithms already known in the literature or whether they were developed by the authors.

Author Response

(The authors gave the same response as above.)

Round 2

Reviewer 1 Report

The authors have answered the question properly in the revised paper. I would like to recommend it to be published in Sensors